# Characterization of Antimicrobial, Antioxidant, and Leishmanicidal Activities of Schiff Base Derivatives of 4-Aminoantipyrine

**DOI:** 10.3390/molecules24152696

**Published:** 2019-07-24

**Authors:** Rommy Teran, Rommel Guevara, Jessica Mora, Lizeth Dobronski, Olalla Barreiro-Costa, Timo Beske, Jorge Pérez-Barrera, Ramiro Araya-Maturana, Patricio Rojas-Silva, Ana Poveda, Jorge Heredia-Moya

**Affiliations:** 1Facultad de Ciencias Químicas, Universidad Central del Ecuador, Quito 170521, Ecuador; 2Instituto de Investigación en Salud Pública y Zoonosis-CIZ, Universidad Central del Ecuador, Quito 170521, Ecuador; 3Centro de Investigación Traslacional, Universidad De Las Américas, Quito 170503, Ecuador; 4Centro de Investigación Biomédica (CENBIO), Facultad de Ciencias de la Salud Eugenio Espejo, Universidad UTE, Quito 170527, Ecuador; 5Facultad de Medicina Veterinaria, Universidad Central del Ecuador, Quito 170521, Ecuador; 6Instituto de Química de Recursos Naturales, Programa de Investigación Asociativa en Cáncer Gástrico (PIA-CG), Universidad de Talca, Talca 3460000, Chile

**Keywords:** schiff base synthesis, Leishmania, antimicrobial activity, cytotoxicity

## Abstract

Our main interest is the characterization of compounds to support the development of alternatives to currently marketed drugs that are losing effectiveness due to the development of resistance. Schiff bases are promising biologically interesting compounds having a wide range of pharmaceutical properties, including anti-inflammatory, antipyretic, and antimicrobial activities, among others. In this work, we have synthesized 12 Schiff base derivatives of 4-aminoantipyrine. In vitro antimicrobial, antioxidant, and cytotoxicity properties are analyzed, as well as in silico predictive adsorption, distribution, metabolism, and excretion (ADME) and bioactivity scores. Results identify two potential Schiff bases: one effective against *E. faecalis* and the other with antioxidant activity. Both have reasonable ADME scores and provides a scaffold for developing more effective compounds in the future. Initial studies are usually limited to laboratory in vitro approaches, and following these initial studies, much research is needed before a drug can reach the clinic. Nevertheless, these laboratory approaches are mandatory and constitute a first filter to discriminate among potential drug candidates and chemical compounds that should be discarded.

## 1. Introduction

In the last decades, antibiotic resistances have been described in several microorganisms, including those having multiresistance phenotypes [1,2,3,4]. This is an alarming situation, as noted by the World Health Organization, and many researchers have focused on the development of new therapeutic alternatives. Oldfield and Feng have proposed different strategies to develop a new generation of antibiotics (resistance-resistant antibiotics) that should have the characteristic of acting against multiple targets simultaneously [5]. Alternatively, other authors have suggested the development of new antibiotic classes with novel chemical properties not present in existing classes of antibiotics [6]. In this context, Schiff bases offer an alternative that has not been adequately explored.

Schiff bases have received much attention due to their wide applications in several fields, ranging from industrial uses such as advanced nanomaterials [7] to chemotherapeutics and new drug development [8,9,10,11]. However, even regarding the chemotherapeutic applications, the reported roles are very divergent, sometimes with proposed roles that could seem contradictory. For example, antioxidant activity has been ascribed to Schiff bases, acting as cell-protective agents [12,13,14,15], but antimicrobial [16,17], antitumor [18,19], analgesic [12], anti-inflammatory [9], and antidiabetic activities [8,18] have also been reported. A detailed review covering the patented therapeutics applications of Schiff bases has been published by Khan and collaborators [18]. These diversities of functions could rely on the wide variety of Schiff base compounds that can interfere with biological activities. The open question is what is the mechanism(s) of action by which they operate. Recent publications suggest that different groups of Schiff bases can exert actions through different mechanisms. For example, some compounds have a potential application as antitumor drugs [18], but different molecular modes of action could explain this property. While some Schiff bases are able to intercalate into the DNA helix [6], others display DNA cleavage activity [20,21]. It is not possible to rule out the possibility that in some compounds, both activities are linked. Also, this DNA-damaging activity could explain antimicrobial activities in some cases.

Other Schiff bases seem to act by inhibiting specific targets. Some of the most prominent examples are the patented inhibitors specific for β-secretase, an enzyme present in aberrantly high levels in Alzheimer’s patients [22]; Schiff base copper complexes that target and inhibit reversibly caspase enzymes involved in apoptosis [23]; or Schiff bases of thiazoles specific for urease, an enzyme present in bacteria, fungi, and plants [24] (all of them reviewed by Khan and collaborators [18]).

However, the broad range of described effects of these compounds make it reasonable to think that a more generalized mechanism may be involved. The role of Schiff bases in a generalized process, the in vivo nonenzymatic glycation of macromolecules, is very interesting. These reactions take place between a reducing sugar and a macromolecule—DNA, lipid, or protein—and generate advanced glycation end products (AGEs) [8,25] through a Schiff base intermediary. The accumulation in blood of AGE products, closely related to oxidative stress, triggers many health complications associated with diseases such as ageing, diabetes, Alzheimer’s, and cancer, among others [8,25,26]. A class of Schiff bases has been described as agents with antiglycation activity, being very promising for prevention of late diabetic complications [27]. This suggests that some Schiff bases can interfere with AGE production, possibly indicating a generalized mode of action that can explain, at least partially, the wide roles and activities of these compounds. Protein glycation leads to crosslinking and protein aggregation, which indeed constitute a source of oxidative species. Some proteins suffering massive glycation are immunoglobulins, apolipoprotein, fibrinogen, and transferrin, which have been associated with several disorders [25]. It is possible that the antioxidant activity described for some Schiff bases [13,18] is related to the prevention of the protein glycation processes, but to our knowledge, nothing has been published regarding this; however, this correlation has been reported for some flavonoids inhibiting the formation of both initial and advanced stages of Maillard reaction in tissue protein sources [28]. Other mechanisms could operate by trapping of carbonyl compounds, similar to that reported with OPB-9195, a known inhibitor of in vitro formation of AGEs [29].

The diversified Schiff base structures and their different mechanisms of action can explain the wide variety of biological activities described for these compounds that deserve more attention as promising candidates for therapies are being more and more delineated. Some of the Schiff bases reported with interesting bioactivities are those derived from 4-aminoantipyrine, and early reports have shown that these compounds have also important antioxidant properties [14], as well as antifungal and antibacterial activity against several microorganisms [30,31,32,33]. However, most of these studies were performed by antibiograms. Here, we report the synthesis of a series of Schiff base derivatives of 4-aminoantipyrine [14,30,31,32,34,35,36,37] and their evaluation of antifungal, antibacterial, (inhibitory concentration of 50% of population (IC_50_), minimum inhibitory concentration (MIC), and noninhibitory concentration (NIC)), leishmanicidal, and antioxidant activities, which will provide useful information to explain the potential chemotherapeutic use of these compounds.

## 2. Results

### 2.1. Synthesis of Schiff Base of 4-Aminoantipyrine

The Schiff bases **3** were synthesized following the scheme depicted in Figure 1. Compound **1** was reacted with the respective benzaldehydes **2a**–**j** to obtain **3a**–**j**, while **4** and **5** were obtained by reaction with furfural (**6**) and cinnamaldehyde (**7**), respectively. The reactions were performed using ethanol as solvent to give good to excellent yields (Table 1). All compounds were characterized, and all the data obtained agreed with the proposed structures. The ^1^H NMR spectra for **3a**–**j** show a singlet between 9.41 and 9.79 ppm corresponding to the azomethine –CH=N proton, while **5** shows a doublet at 9.40 ppm. These signals are like those reported in the literature, and as expected, the azomethine proton for all compounds with electron-donating groups are shifted to a higher field relative to **3a**, except for **3b**, which shows a signal at 9.70 ppm, possibly due to an intramolecular hydrogen bond with the hydroxyl group.

### 2.2. Antimicrobial Evaluation

The in vitro antifungal and antimicrobial activities were determined by testing the Schiff bases against the fungi *Saccharomyces cerevisiae*, *Candida albicans*, and *Aspergillus niger* and the bacteria *Staphylococcus aureus*, *Enterococcus faecalis* (Gram-positive), *Pseudomonas aeruginosa*, and *Escherichia coli* (Gram-negative). Additionally, the values of minimum inhibitory concentration (MIC), noninhibitory concentration (NIC), and inhibitory concentration of 50% of population (IC_50_) were determined as described by Lambert et al. [38,39].

Since the Schiff bases have a limited solubility in dimethyl sulfoxide (DMSO), first, we tested the toxicity of DMSO against the selected microorganism. According to published data [40], concentrations over 2% DMSO are significantly toxic for *S. cerevisiae*. Drop test assay (Figure 2) revealed that wild-type cells from the BY4741 strain grow with no major difference at DMSO concentrations below 2%, but at concentrations of 3% DMSO or higher, wild-type cells begin to show diminished viability, and the effect is more acute in *sod1* cells. Wild-type cells from W303 background seem to be more resistant, and sensitivity is observed at around 4% DMSO.

More accurate determinations of noninhibitory concentration (NIC) [38] revealed that toxicity ranges from a minimum of 1.55% DMSO (*E. faecalis*) to a maximum of 15.86% (*E. coli*) (Table 2).

Considering the limited solubility of Schiff bases and the DMSO toxicity, we set up our protocol at a maximal DMSO working concentration of 2%. Values of NIC, MIC, and IC_50_ are summarized in Table 3, and all parameters were compared with that of the commercial drugs geneticin (GN) for *S. cerevisiae*, voriconazole (VR) for *A. niger* and *C. albicans*, and ciprofloxacine (CP) for all bacteria studied. It is important to note that although in the literature there appear several reports regarding the activities of some of Schiff bases reported here with these microorganisms [14,30,31,32,33,34,41,42,43,44,45,46,47,48], those tests were done using different methods of evaluation in different strains, when it was reported. Overall, *A. niger* and *S. aureus* are inhibited by Schiff base **3f**; *C. albicans* by **3b** and **3f**; *S. cerevisiae* by **5**; *E. faecalis* by **3a**; and *E. coli* by **3b**, **3d**, **3f**, and **5**. Only *P. aeruginosa* was not sensitive to any of the assayed compounds. Interestingly, different microorganisms are inhibited by different compounds, revealing specific activities against specific microorganisms. The specificity is a property required in drug candidates for applications in pharmacology. Compounds **3c**, **3e**, **3g**, **3h**, **3i**, **3j**, and **4** did not show any activity against the tested microorganisms, and the activity of **3e**, **3f**, and **5** had not been reported previously. Schiff base **3f** is the only compound that showed activity with all bacteria, except *E. faecalis*, and **3d** had the lowest activity (MIC of 10.2 ± 0.1 μg/mL). When the observed MIC concentration was >200 μg/mL or out of the assay range concentration, the drug was considered ineffective (reported as NE, no effect).

Next, we evaluated qualitatively if the inhibitory effect is exerted by killing the microorganism (bactericidal/fungicidal effect) or by preventing growth (bacteriostatic/fungistatic effect). After incubating cells in the presence of different concentration of Schiff bases for 24 h, cells were plated on media free of drug. Results indicated that only compound **5** has a fungicidal mechanism on *S. cerevisiae*, since cells are not able to recover after exposure (see Appendix A). The rest of the Schiff bases assayed showed a bacteriostatic behavior, allowing the growth of microorganism on drug-free media after 24 h of exposure (Appendix A).

Finally, Schiff bases were evaluated against *L. mexicana* parasites in promastigote form by (3-(4,5-Dimethylthiazol-2-yl)-2,5-diphenyltetrazolium bromide (MTT) assay. The results are shown in Table 3. Some compounds were inactive or too weak (a cutoff value of IC_50_ > 200 µM was considered), but others showed leishmanicidal activity to different extents, particularly **5** and **3f**, with the latter being the most powerful. Pentamidine and amphotericin were included as controls.

In conclusion, Schiff bases **3f** and, to a lesser extent, **5** show a wide range of activities against several bacteria. Also, **3a**, **3b**, and **3d** are interesting candidates, with narrow but specific activity. Compounds **3f** and **5** showed the best activity against parasites. So, we will focus our next efforts in evaluation of these compounds.

### 2.3. Evaluation of Antioxidant/Oxidative Activity

All the synthesized compounds were evaluated for antioxidant activity by the *N*,*N*-diphenyl-*N*′-picrylhydrazyl (DPPH) assay (Table 4). All compounds evaluated showed lower radical scavenging activity than the standards ascorbic acid (entry 13), quercetin (entry 14), and caffeic acid (entry 15). In general, the presence of hydroxyl groups in the *para* position of the aromatic ring (**3d**, **3i**, and **3j**) confers a better radical scavenging activity compared to those with hydroxyl groups in other positions or with other electron-donating groups, with the exception of **3h**, which contains one dimethylamino group (entry 8) that also can stabilize the radical by resonance. Compounds **3d**, **3h**, **3i**, and **3j** possess the best antioxidant activities, comparable to the standards, but only **3d** presents activity against bacteria (*E. coli*), while **3f** and **5** are effective against leishmania promastigotes.

Next, we wanted to evaluate the oxidative activity. To do this, drop test assays were performed with wild-type (W303 and BY4741 backgrounds) and *sod1* (BY4741 background) yeast cells in the presence of 200 µg/mL of the analyzed Schiff bases (Figure 3). The SOD1 gene encodes superoxide dismutase 1, an enzyme with a major role in detoxifying ROS species generated during metabolism or because of environmental exposure. s*od1* mutant cells are unprotected against oxidative agents, decreasing survival in their presence. Figure 3A shows the relevant results obtained with drugs **3b** and **5**, where *sod1* cells are sensitive to these compounds compared to the control (no drug, DMSO), after 48 h of incubation at 25 °C. Also, wild-type cells are shown as control, unaffected by the Schiff bases. Figure 3B and Table 5 show the graphic representation of optical density (OD600) of wild and *sod1* cells exposed to increasing concentrations of compounds **3b** and **5**, and the MIC, IC_50_, and NIC values obtained for these compounds, respectively.

These results reveal that **3b** and **5** act by an oxidative mechanism, in accordance with the results obtained in the DPPH assay, which detected no significant antioxidant activity of these compounds.

### 2.4. Toxicity in Mammalian Cell Lines

The specificity and selectivity for the pathogen target is a key point for any potential drug. To evaluate these, we determined the cytotoxicity (% of living cells) of mammal macrophage cells (murine RAW 264.7, ATCC TIB 71) after exposure to Schiff bases; see Table 6. Cytotoxicity was determined at 400 μM of each Schiff base, except for those marked with an asterisk (*), which were assayed at 300 μM because of the limited solubility inherent to these compounds. Compounds **3d**, **3f**, **3h***, **3j**, and **5** showed high cytotoxicity with ≤20% of cells surviving, with **3f** and **5** being the most cytotoxic, thus making them useless for pharmacological purposes, at least in this simple formulation (chemical compound directly in solution). Cytotoxicity data restrain the potential application to **3a** and **3i**, the only two compounds producing viability of >25% at 0.1165 and 0.1293 mg/mL, respectively. A control with saponine was included, giving an IC_50_ value of 0.1400 mg/mL. **3a** and **3i** are effective against *E. faecalis* and as an antioxidant, respectively. Overall, these results are an early filter indicating a potential application for compounds **3a** and **3i**, which deserve to be analyzed more deeply.

### 2.5. Evaluation of Pharmacokinetic and Drug-Like Properties

To evaluate the drug-likeness of a compound, Lipiniski’s rule of five (ROF) is usually used. This rule states that good absorption or permeation are more likely when: (a) there are no more than five hydrogen bond donors (HBD); (b) no more than 10 hydrogen bond acceptors (HBA); (c) formula weight less than 500; and (d) *n*-octanol/water partition coefficient (log P) is less than 5. Two or more violations of ROF suggest the probability of problems in bioavailability [49]. One of the big problems with the development of new drugs is that they usually fail before reaching the clinics due to poor pharmacokinetics, and these physicochemical drug descriptors provide a useful tool for evaluating drug activity. These descriptors were calculated for all the synthesized compounds **3a**–**j**, **4**, and **5** using the Osiris DataWarrior software (Table 7) [50]. Additionally, the bioactivity scores of all compounds were calculated using the Molinspiration cheminformatics software (Table 8), and all parameters were compared with those of the standard drugs voriconazole (VR) and ciprofloxacin (CP).

Calculations of the physicochemical drug descriptors (see Table 7) show that molecular weights of all synthesized compounds are below 500; HBD and HBA values are 4–7 and 0–2, respectively; logP and logS values were below 2.2 and above −3.70, respectively; and all compounds have a low number of nrotb (3–5); so, a very low conformational flexibility and good oral bioavailability are expected. According to this, all Schiff bases meet Lipinski´s rule of five. Additionally, all compounds show permeability to cell membranes, and half of them (**3a**–**c**, **3h**, **4**, and **5**) show PSA values from 35.91 to 56.14 Å^2^, suggesting good permeability at the blood–brain barrier.

The evaluation of toxicity risk assessment is also important prior to structural drug design, and many potential drugs fail to reach the clinical stage because of ADME toxicity issues. The toxicity risk predictor locates fragments within a molecule that constitute a potential toxicity risk for one of four major toxicity classes: mutagenicity, tumorigenicity, reproductive effect, and irritant effect. The results are displayed, regarding the category specified, as high, low, or none as risk for a compound to possess a toxic effect. The result show that all the compounds have a high risk to be mutagenic and only **3d** and **3h** show a low and high risk, respectively, to be tumorigenic. In the other categories, none of the compounds showed effect.

Drug-likeness (DL) of a compound may be defined as a complex balance of various molecular properties and structural features which determine whether a particular molecule is similar to known drugs [33]. Considering that most drugs have drug-likeness with a positive value, it is desirable that new candidate stay in the positive range. Osiris DataWarrior calculations show that all nitro compounds, **3f**, and **3g** have negative values, while other compounds have values higher than 4.08, a value similar to that observed for voriconazole. The drug score index (DS) is a parameter that combines drug-likeness, cLogP, logS, molecular weight, and toxicity risks in a single descriptor that is very useful for determining whether a compound can be qualified as a potential drug. According to established criteria, values closer to 1 indicate a higher probability of a compound being a future drug candidate. All the evaluated compounds exhibited poor to moderate scores (0.30 to 0.54), **4** being the compound with the highest value. Finally, the bioactivity scores of all compounds for being a G protein-coupled receptor (GPCR) ligand, ion channel modulator, kinase inhibitor, nuclear receptor ligand, protease inhibitor, and enzyme inhibitor were calculated and are shown in Table 8. A high score implies a higher probability for a molecule to be biologically active. For many organic molecules, the prediction is as follows: if the bioactivity score is more than 0.00, it is active; if −0.50 to 0.00, moderately active; and if less than −0.50, inactive. According to this, all compounds in this study are predicted to be inactive as GPCR ligands, ion channel modulators, nuclear receptor ligands, and protease inhibitors, and only **3j** shows a moderate activity as a kinase inhibitor, while **3b**, **3d**, **3e**, **3h**, **3i**, and **3j** show similar activity as enzyme inhibitors. Both standard drugs show moderate to high activity against all bioactivities evaluated. These results suggest that the mechanism of action would be unspecific and/or generalized.

## 3. Discussion

In this study, we synthesized a series of Schiff base derivatives of 4-aminoantipyrine and tested their cytotoxic properties against selected bacteria, fungi, and leishmania parasites, as well their antioxidant/oxidative activities. Finally, we analyzed their properties regarding their suitability for therapeutic uses.

Schiff bases are an emerging alternative for disease treatments. In the last decades, many types of resistance have been described in bacteria, generating the need for new drugs for dealing with the diseases caused by microbes [3,4]. In addition, the prevalence of cancer and metabolic diseases such as diabetes are predicted to rapidly grow in the next years [51,52]. For many of these diseases, there are still no effective therapies. Schiff bases have shown a wide range of properties that make them promising candidates as alternative drugs, but they have not been adequately characterized. There have been described—and some patented—Schiff bases that are effective against microorganisms (bacteria, virus, fungi, parasites) or against cancer, with antiglycation, antioxidant, or anti-inflammatory activities, among others [15,18]. A major question is how these compounds act against several targets and display different properties. The most reliable answer is that due to the vast variety of compounds, there should be many modes of action.

Our results indicate that Schiff bases **3a**–**b**, **3d**, **3f**, and **5** show antimicrobial activities with moderate inhibitory values (Table 3). Evaluation of scavenging activity reveals antioxidant properties for **3d** and **3h**–**j**, while compounds **3b**, **3f**, and **5** possess oxidative activity. Overall, according to the in silico calculations, Schiff bases present good scores for the parameters evaluated, such as bioavailability, and have membrane permeability similar to commercial drugs (DS/DL), except **3d** and **3f**–**h**, which have poor scores for DS (<0.5), DL, and risk of tumorigenicity. The main problem with the low DL score of **3f** is the presence of the nitro group in the structure, with is related to the toxicity of some drugs [53]. Actually, drug discovery pipelines for the ‘neglected diseases’ are now heavily populated with nitro compounds [54,55], so this compound should not be discarded. Also, cytotoxicity is significant in mouse macrophages. These analyses limit compounds **3a** and **3i** to be the ones with the most promising potential applications (summarized in Figure 4), with some considerations discussed below. Compounds **3c**, **3e**, **3g**, and **4** do not present any kind of activity and were discarded from further analysis. It deserves to be discussed that compound **3g** is equivalent to compound **9** previously described by Thirunarayanan and collaborators [30]. The authors describe microorganism sensitivity by the Kirby–Bauer method with comparison to the commercial drugs used as controls, namely ciprofloxacin for bacteria and miconazol for fungi. They reported significant activity against *B. subtilis*, *M. luteus*, *S. aureus*, *E. coli*, *P. aeruginosa*, and the fungi *A. niger* and *T. viride*. In our hands, only *S. cerevisiae* presents a relative sensitivity to Schiff base **3g**, but for the rest of the tested microorganisms, the MIC values obtained were too high (>200 μg/mL). These two techniques are not directly comparable, since the Kirby–Bauer method is qualitative and MIC is quantitative, but two possible explanations for these differences are as follows: First, Thirunarayanan and collaborators did not report the disk potency (μg) used in the assay; and second, the origin of the strains used in their work is not indicated and apparently are not from the American Type Culture Collection (ATCC). Together, this could explain the differences obtained with the same compound. Nevertheless, we should omit compound **3g** from next analyses, since we are not able to describe any significant activity.

Compounds **3b**, **3f**, and **5** show both antimicrobial and oxidative activity, opening the possibility that the antimicrobial effect is a consequence of its oxidative activity. Since the effect is specific (**3b** for *C. albicans* and *E. coli*; **3f** for *C. albicans*, *A. niger*, *S. cerevisiae*, *E. coli*, *S. aureus*, and *L. mexicana*; **5** for *S. cerevisiae*, *E. coli*, and *L. mexicana*), we cannot rule out that these compounds are oxidizing specific targets. This is consistent with the bioactivity analyses, at least for **3b**, indicating that it could be acting as an enzyme inhibitor (Table 8). However, due to the difference observed in the time-response experiments, the action mechanism of both compounds could be different. Additionally, **3a** and **3d** showed antimicrobial activity against *E. faecalis* and *E. coli*, respectively.

The analysis of the structure–activity of the Schiff bases shows that **5** presents an extension in the imine conjugation and the aromatic ring does not possess any substituent, while **3f** has an *ortho* hydroxyl group (electron-donating group) and a *meta* nitro group (electron-withdrawing group), and it is evident there is a synergistic effect between both substituents, because when only the hydroxyl group (**3b**) or only the nitro group (**3g**) is present, the activity is limited to *C. albicans* and *E. coli* in the first case, and no activity observed in **3g**, while no activity against leishmania was observed for either compound. The hydroxyl group in the *ortho* position can form an intramolecular hydrogen bond with the imine nitrogen [56], which, due to the inductive and resonance effect of the nitro group, favors the keto–enol tautomerism to generate the keto tautomer which could be, in part, responsible for the enhancement of the observed antimicrobial and leishmanicidal activities.

Toxicity was assayed in RAW cells (Table 6). Results indicate that compounds **3a** and **3i** are the least toxic in our system. Although it is true that other systems should be tested, such as primary macrophages isolated from mice or human macrophages cell lines, the results with RAW cells are a sensitive indication of toxicity that should be considered before continuing any new experimental design. Indeed, **3a** and **3i** are effective against *E. faecalis* and as antioxidant, respectively. *E. faecalis* is an opportunistic pathogen highly associated with resistance development in the last decades. This bacterial infection is common in hospital environments, presenting such clinical manifestations as urinary tract infections, bacteremia, and endocarditis, among others [57].

Regarding compounds **3d** and **3h**–**j**, their potential application is related to their antioxidant activity, and thus as cell-protective agents. In accordance with our results, antioxidant activity for compounds **3i**–**j** has been previously described [14], along with a very low inhibition of RAW cell viability. These compounds have similar activities as the controls: ascorbic acid, quercetin, or caffeic acid (Table 4). Nevertheless, potential applications of **3d** and **3h** are not very promising, as the ADME test gave poor scores for several of the parameters (Table 7). Neither is **3j** a good candidate, as the percentage of viable RAW cells was compromised in our experiments (Table 6). So, only **3a** seems to be a reliable option for an antioxidant activity approach.

The biggest limitation of these compounds is that all of them seem to present mutagenicity characteristics, according to our predictive analyses for ADME toxicity issues (Table 7). If so, this prediction should be tested experimentally; it would be interesting to exploit these compounds to identify chemical varieties that keep or improve their useful properties, while diminishing the associated mutagenicity. Nevertheless, a possible explanation is that compounds **3a**, **3b**, **3d**, **3f**, and **5** exert their antimicrobial and/or leishmanicidal activities through a mutagenic mechanism, but this could not be the main mechanism as none of the other compounds show this activity, despite reported mutagenicity. Although this is not a desired property for antimicrobial compounds, this observation opens the question as to whether these compounds could be used in cancer treatment.

It is recognized that most anticancer drugs are potentially hazardous substances, since they are mutagenic, teratogenic, and/or carcinogenic [58] and as they exert their activities probably due to these properties. Some of them have been classified as carcinogenic by the International Agency for Research on Cancer (IARC), and the in silico prediction of their ADME properties show poor scores in some categories (Appendix A). For example, cyclophosphamide, a drug used in the treatment of chronic lymphocytic leukemia, lymphomas, soft tissue and osteogenic sarcomas, and solid tumors, is in group 1 (carcinogenic to humans) according to IARC, and according to the calculations, presents a high risk to be mutagenic, tumorigenic, and have reproductive effects. Considering this, it cannot be ruled out that Schiff bases could have anticancer properties. Indeed, a review of the literature describes some Schiff bases with associated anticancer activity [32]. However, more studies should be done to determine if these compounds are able to discriminate between normal mammalian cells and certain kinds of pathogens or cancer cells [18,19], or if, in contrast, they are generalized mutagens useless for therapeutics.

The toxicity of drugs can be greatly controlled using nanoformulations (nanoparticles, nanotubes, liposomes, etc.) [41,42]. These systems enable specific selection of the desired target and delivery of the drug locally and in a controlled way, reducing the side effects generated by conventional drug delivery systems that usually require high dosage. So, the choice of the system will depend on the final purpose of the formulation. Nanomedicine offers new approaches to drug design, improving the specificity and limiting the undesired toxicity. Major efforts have been focusing on developing nanocarriers to fight against cancer. Main limitations of chemotherapy are multidrug resistance and nonspecific tissue targeting, generating serious side effects that compromise the patient’s health. Nanoformulations of chemotherapeutic agents enhance the selective delivery, diminishing the toxicity. Despite the associated benefits, only a few nanoformulations for chemotherapy have been approved by the FDA and are currently marketed. Nevertheless, many research projects and clinical trials are studying different nanoparticles, disclosing them as an emerging therapeutic approach [43,52]. Nanoparticles are not only useful in chemotherapy; many potential applications against infectious microorganisms and brain or metabolic diseases are emerging [41,44].

The work here presented is a first approach to discriminate among compounds to determine those which have application potential and those which are useless (have no biological activity or intrinsic toxicity, etc.). Additional efforts to characterize and improve Schiff base properties and toxicity should be done by the scientific community to uncover the real uses and applications of these compounds in therapy as antimicrobial agents or otherwise, such as anti-inflammatory, antipyretic, antitumor, or antidiabetic agents.

## 4. Materials and Methods

### 4.1. General Procedures

All solvents and reagents were from Sigma Aldrich (St. Louis, MO, USA) and used without further purification. All melting points are uncorrected and were determined on a Büchi Melting Point M-560 (Büchi Labortechnik AG, Flawil, Switzerland). Infrared spectra (KBr discs) were recorded on a Perkin Elmer RX1 infrared spectrophotometer (PerkinElmer, Waltham, MA, US); wave numbers are reported in cm^−1^. NMR spectra were recorded at 298 K on a Bruker Advance 500 MHz spectrometer (Bruker BioSpin GmbH, Rheinstetten, Germany) equipped with a z-gradient, triple-resonance (^1^H, ^13^C, ^15^N) cryoprobe using CDCl_3_ as solvent or in a Bruker AVANCE DRX 300 spectrometer (Bruker BioSpin GmbH, Rheinstetten, Germany) using DMSO-*d_6_* as solvent. Chemical shifts are expressed in ppm with TMS as an internal reference. Accurate mass data were obtained using a Waters model LCT Premiere time-of-flight (TOF) mass spectrometer (Waltham, MA, US). Absorbance was measured at different wavelength on a Multi-Mode Microplate Reader Synergy H1 (Biotek, VT, US) or HT BioTek spectrophotometer (Biotek, VT, US). Reactions were monitored by TLC on pre-coated silica gel aluminum plates, Kieselgel 60 F254 from Merck (Darmstadt, Germany) using ethyl acetate/hexane mixtures as a solvent and compounds were visualized by UV lamp (UVP, Upland, CA, US). The reported yields are for the purified material and are not optimized. The IC_50_ values were calculated in GraphPad Prism 7.02 (GraphPad Software, San Diego, US) The results are given as a mean ± standard deviation (SD) of experiments done in triplicate.

### 4.2. Synthesis of Schiff Base Derivatives (**3a**–**j**, **4**, **5**)

The synthesis of the Schiff base derivatives **3a**–**j** was adapted from a procedure reported by Alam and coworkers [14]. To an ethanol solution (10 mL) of 4-amino-1,5-dimethyl-2-phenylpyrazol-3-one (**1**) (1.0 g, 4.92 mmol) was added an ethanol solution (10 mL) of substituted benzaldehyde (4.92 mmol), and the mixture was refluxed for 6 h. The progress of the reaction was monitored by TLC. The precipitates formed were collected by filtration, purified by recrystallization with ethanol, and then dried under vacuum to produce the pure compounds. Spectra analyses can be consulted in the Appendix A.

4-Benzylideneamino-1,5-dimethyl-2-phenyl-1*H*-pyrazol-3(2*H*)-one (**3a**): yield 77% as white crystals; m.p. 176.8–177.2 °C (Lit [14] 178.3 °C); ^1^H-NMR (300 MHz, DMSO-*d*_6_) δ 9.59 (s, 1 H), 7.81 (dd, *J* = 7.3, 2.1 Hz, 2 H), 7.49–7.57 (m, 2 H), 7.41–7.48 (m, 3 H), 7.34–7.41 (m, 3 H), 3.18 (s, 3 H), 2.46 (s, 3 H).

4-(2-Hydroxybenzylideneamino)-1,5-dimethyl-2-phenyl-1*H*-pyrazol-3(2*H*)-one (**3b**): yield 94% as yellow crystals; m.p. 201.0-201.5 °C (Lit [34] 199 °C); ^1^H-NMR (300 MHz, DMSO-*d*_6_) δ 12.94 (s, 1 H), 9.70 (s, 1 H), 7.51–7.59 (m, 2 H), 7.45–7.50 (m, 1 H), 7.36–7.44 (m, 3 H), 7.28–7.35 (m, 1 H), 6.88–6.95 (m, 2 H), 3.21 (s, 3 H), 2.41 (s, 3 H).

4-(4-Methoxybenzylideneamino)-1,5-dimethyl-2-phenyl-1*H*-pyrazol-3(2*H*)-one (**3c**): yield 86% as white crystals; m.p. 169.8–170.3 °C (Lit [32] 171–173 °C); ^1^H-NMR (300 MHz, DMSO-*d*_6_) δ 9.52 (s, 1 H), 7.76 (d, *J* = 8.8 Hz, 2 H), 7.57–7.49 (m, 2 H), 7.41–7.33 (m, 3 H), 7.02 (d, *J* = 8.8 Hz, 2 H), 3.81 (s, 3 H), 3.15 (s, 3 H), 2.44 (s, 3 H).

4-(4-Hydroxy-3-methoxybenzylideneamino)-1,5-dimethyl-2-phenyl-1*H*-pyrazol-3(2*H*)-one (**3d**): yield 94% as white crystals; m.p. 210.6–210.9 °C (Lit [32] 208–210 °C); ^1^H-NMR (300 MHz, DMSO-*d*_6_) δ 9.46 (s, 1 H), 7.48–7.57 (m, 2 H), 7.32–7.44 (m, 4 H), 7.19 (dd, *J* = 8.2, 1.7 Hz, 1 H), 6.84 (d, *J* = 8.2 Hz, 1 H), 3.84 (s, 3 H), 3.14 (s, 3 H), 2.44 (s, 3 H).

4-(3-Hydroxy-4-methoxybenzylideneamino)-1,5-dimethyl-2-phenyl-1*H*-pyrazol-3(2*H*)-one (**3e**): yield**:** 93.4% as white crystals; m.p. 246.8–247.2 °C; ^1^H-NMR (500 MHz, *CDCl_3_*) δ 9.66 (s, 1 H), 7.61 (d, *J* = 1.3 Hz, 1 H), 7.48 (dd, *J* = 8.2, 7.7 Hz, 2 H), 7.42 (dd, *J* = 8.5, 1.4 Hz, 2 H), 7.32 (tt, *J* = 7.2, 1.2 Hz, 1 H), 7.29 (dd, *J* = 8.2, 1.7 Hz, 1 H), 6.89 (d, *J* = 8.2 Hz, 1 H), 5.71 (br. s, 1 H), 3.94 (s, 3 H), 3.14 (s, 3 H), 2.49 (s, 3 H); ^13^C NMR (126 MHz, *CDCl_3_*) δ 161.0, 156.9, 151.7, 145.8, 134.9, 129.2, 129.0, 126.8, 125.9, 124.3, 122.8, 114.1, 112.1, 110.2, 56.0, 35.9, 10.2; IR (cm^−1^) (KBr discs) 3096.15, 2958.95, 1608.34, 1573.63, 1257.36, 1020.16, 776.21; HRMS (TOF ES+) *m*/*z* calcd for C_19_H_20_N_3_O_3_ (M+H)^+^: 338.1505; found: 338.1506.

4-(2-Hydroxy-5-nitrobenzylideneamino)-1,5-dimethyl-2-phenyl-1*H*-pyrazol-3(2*H*)-one (**3f**): yield 84.6% as yellow crystals; m.p. 204.9–205.6 °C (Lit [36] 205.3–205.9 °C); ^1^H-NMR (300 MHz, DMSO-*d*_6_) δ 13.73 (br. s., 1 H), 9.79 (s, 1 H), 8.52–8.58 (m, 1 H), 8.18 (dd, *J* = 9.0, 2.8 Hz, 1 H), 7.56 (t, *J* = 7.7 Hz, 2 H), 7.35–7.46 (m, 3 H), 7.10 (d, *J* = 9.2 Hz, 1 H), 3.26 (s, 3 H), 2.45 (s, 3 H).

4-(3-Nitrobenzylideneamino)-1,5-dimethyl-2-phenyl-1*H*-pyrazol-3(2*H*)-one (**3g**): yield 95.4% as light orange crystals; m.p. 219.7–220.1 °C (Lit [30] 218–219 °C); ^1^H-NMR (300 MHz, DMSO-*d*_6_) δ 9.68 (s, 1 H), 8.59–8.63 (m, 1 H), 8.19–8.28 (m, 2 H), 7.74 (t, *J* = 8.0 Hz, 1 H), 7.51–7.59 (m, 2 H), 7.35–7.44 (m, 3 H), 3.24 (s, 3 H).

4-(4-Dimethylaminobenzylideneamino)-1,5-dimethyl-2-phenyl-1*H*-pyrazol-3(2*H*)-one (**3h**): yield 94.5% as yellow crystals; m.p. 220.4–221.4 °C (Lit [37] 223–224 °C); ^1^H-NMR (300 MHz, DMSO-*d*_6_) δ 9.43 (s, 1 H), 7.62 (d, *J* = 8.8 Hz, 2 H), 7.55–7.48 (m, 2 H), 7.40–7.30 (m, 3 H), 6.75 (d, *J* = 8.8 Hz, 2 H), 3.11 (s, 3 H), 2.98 (s, 6 H), 2.41 (s, 3 H).

4-(3,4-Dihydroxybenzylideneamino)-1,5-dimethyl-2-phenyl-1*H*-pyrazol-3(2*H*)-one (**3i**): yield 97.4% as cream powder; m.p. 279.2–280.5 °C (Lit [14] 287.2 °C); ^1^H-NMR (300 MHz, DMSO-*d*_6_) δ 9.41 (br. s., 1 H), 9.38 (s, 1 H), 9.19 (br. s., 1 H), 7.52 (dd, *J* = 8.2, 7.4 Hz, 2 H), 7.32–7.38 (m, 3 H), 7.29 (d, *J* = 1.9 Hz, 1 H), 7.02 (dd, *J* = 8.1, 1.9 Hz, 1 H), 6.78 (d, *J* = 8.1 Hz, 1 H), 3.12 (s, 3 H), 2.41 (s, 3 H).

4-(4-Hydroxy-3,5-dimethoxybenzylideneamino)-1,5-dimethyl-2-phenyl-1*H*-pyrazol-3(2*H*)-one (**3j**): yield 90.5% as white crystals; m.p. 259.2–260.7 °C (Lit [14] 258.9 °C); ^1^H-NMR (300 MHz, DMSO-*d*_6_) δ 9.46 (s, 1 H), 8.88 (br. s., 1 H), 7.53 (dd, *J* = 8.5, 7.1 Hz, 2 H), 7.41–7.33 (m, 3 H), 7.10 (s, 2 H), 3.83 (s, 6 H), 3.14 (s, 3 H), 2.45 (s, 3 H).

4-(2-furfurilideneamino)-1,5-dimethyl-2-phenyl-1*H*-pyrazol-3(2*H*)-one (**4**): yield 85.6% as dark yellow crystals; m.p. 212.7–213.7 °C (Lit [45] 206 °C); ^1^H-NMR (300 MHz, DMSO-*d*_6_) δ 9.43 (s, 1 H), 7.84 (d, *J* = 1.7 Hz, 1 H), 7.53 (dd, *J* = 8.1, 7.7 Hz, 2 H), 7.41 - 7.33 (m, 3 H), 6.96 (d, *J* = 3.4 Hz, 1 H), 6.64 (dd, *J* = 3.4, 1.8 Hz, 1 H), 3.17 (s, 3 H), 2.40 (s, 3 H).

**4**-Phenylallylideneamino-1,5-dimethyl-2-phenyl-1*H*-pyrazol-3(2*H*)-one (**5**); yield 90% as yellow crystals; m.p. 165.5–165.9 °C (Lit [37] 164–165 °C); ^1^H-NMR (300 MHz, DMSO-*d*_6_) δ 9.40 (d, *J* = 8.2 Hz, 1 H), 7.64 (dd, *J* = 8.3, 1.4 Hz, 2 H), 7.57–7.49 (m, 2 H), 7.43–7.29 (m, 6 H), 7.11 (d, *J* = 16.1 Hz, 1 H), 7.01 (dd, *J* = 16.1, 8.3 Hz, 1 H), 3.17 (s, 3 H), 2.39 (s, 3 H).

### 4.3. Evaluation of Antimicrobial Activity

The antimicrobial activities of the synthesized compounds were tested against the Gram-positive bacteria *Staphylococcus aureus* ATCC 29213 and *Enterococcus faecalis* ATCC 29212, the Gram-negative bacteria *Escherichia coli* ATCC 25922 and *P. aeruginosa* ATCC 27853, and the fungal strains *Candida albicans* ATCC 10231, *Aspergillus niger* ATCC 6275, and *S. cerevisiae* (backgrounds W303 and BY4741), using the microdilution method [46,47].

The bacterial inoculum was prepared by emulsifying overnight colonies from an agar medium in broth (Mueller–Hinton broth). The suspensions were adjusted to a final organism density of 5 × 10^5^ colony former units (cfu)/mL. The suspensions of *C. albicans* were made in saline solution and adjusted to an optical density of 0.155 at 530 nm. The emulsifying of *A. niger* cultures was done by pouring 5 mL of saline and 0.1% of polysorbate 80 over the Petri dishes. Then, 100 µL of this suspension were diluted in 5 mL of saline and adjusted to an optical density of 0.1 at 530 nm. For *S. cerevisiae*, fresh exponential YPAD (yeast extract 1%, peptone 2%, adenine 0.004%, dextran 2%) cultures were adjusted to an optical density of 0.5 at 600 nm, and then diluted to the equivalent of 1 × 10^4^ cells/mL (10^−3^ dilution). Stock solutions of Schiff bases were prepared by dissolving them in DMSO at 10 mg/mL. For controls, ciprofloxacin was used for all bacteria tested, voriconazol for *A. niger* and *C. albicans*, and geneticin for *S. cerevisiae*. The range of concentrations used was empirically selected for each organisms and the Schiff base. Drug sensitivities were assayed by serial dilutions in 96-well plates as previously described [39,46,47]. A row with media and cells but without drug was the positive control. The blank was done by preparing a row with media and drugs, but without cells. Plates were fitted with a tight lid before incubation to prevent desiccation. Plates were incubated at 37 °C in air for 24 h for bacterial cultures and 35 °C for 48 h for fungi. The turbidity was recorded at 600 nm for bacteria and 530 nm for fungi in a plate reader (Synergy H1, Biotek), and calculations of MIC (minimal inhibitory concentration), NIC (noninhibitory concentration), and IC_50_ (inhibitory concentration for 50% of the population) were done with GraphPad Prism [39]. The minimum bactericidal concentration (MBC) and fungicidal concentration (MFC) were determined by spreading 100 µL of culture from each well onto a plate (Mueller–Hinton agar for bacteria, Sabouraud dextrose agar for fungi except *S. cerevisiae*, and YPAD for *S. cerevisiae*). MBC and MFC values represented the lowest concentration of a compound that completely inhibited growth.

### 4.4. Evaluation of Leishmanicidal Activity

Leishmanicidal activities of all Schiff bases were evaluated by measuring promastigote mitochondrial activity using MTT colorimetric assay as described previously [57]. In this study, promastigotes from *L. mexicana* (M379), donated from the Instituto de Medicina Tropical Alexander von Humboldt (Universidad Cayetano Heredia, Peru) were cultured at 24 °C in Schneider’s Drosophila Medium supplemented with 20% fetal bovine serum and 100 IU/mL penicillin + 100 mg/mL streptomycin. The medium was renewed every four days. Parasite density was determined in a Neubauer chamber.

Into each well of a 96-well cell culture dish was dispensed 1 × 10^6^ parasites. A stock solution of each of the Schiff Base was prepared in DMSO, and serial dilutions were added to the parasite suspension, keeping the solvent concentration at 1%. The final volume was 100 µL for each well and triplicate conditions were carried out. Pentamidine treatment (100 µg/mL) and untreated parasites were used as positive control and negative control, respectively. After exposure to the compounds for 48 h in culture medium, 10 µL of a solution of 5 mg/mL MTT dissolved in PBS was added to each well. The plate was incubated at 24 °C for 4 h in darkness. Later, the plate was centrifuged at 4400 rpm for 10 min and the culture medium was then aspirated. Next, 100 µL of DMSO were added into each well to solubilize the formazan and the plate was shaken for 5 min. The colored formazan salt was measured by recording changes in absorbance at 570 nm using a microplate reader, a BioTek Synergy HT spectrophotometer. A reference wavelength of 630 nm was used to subtract background. Optical densities were analyzed as the quantity of formazan is directly proportional to the number of viable parasites. Data were analyzed with the statistical software GraphPad Prism.

### 4.5. Cell Viability

RAW 264.7 cells were maintained in Dulbecco’s Modified Eagle Medium (DMEM) (Gibco, Invitrogen, NY, USA) supplemented with 10% fetal bovine serum (FBS) (Eurobio) and 100 IU/mL penicillin + 100 µg/mL streptomycin (Gibco, Invitrogen) at 37 °C in an atmosphere containing 5% CO_2_. The medium was renewed once a week. The viability was determined using MTT (thiazolyl blue tetrazolium bromide) dye assay as described previously for leishmanicidal activity assessment, with some variations. Here, 5 × 10^4^ cells/well in a final volume of 100 µL were deposited into a 96-well plate in triplicate. Saponin (2 and 4 mg/mL) and untreated cells were used as the positive control and negative control, respectively. Schiff bases were dissolved in DMSO to obtain different serial concentrations (0.01–2000 µM). After 48 h exposure to the compounds, 10 µL/well of MTT (5 mg/mL MTT in PBS) was added, and the plate was incubated at 37 °C for 2 h in darkness. Cells were pelleted by centrifugation at 4400 rpm for 10 minutes and the media was removed. Next, 100 µL/well of DMSO were added and the absorbance at 570 nm was recorded. A reference wavelength of 630 nm was used for background subtraction. 

### 4.6. Drop Test

For the drop test with *S. cerevisiae*, wild-type strain (background W303), and wild type and *sod1* mutant cells (background BY4741) were grown overnight in YPAD media until an optical density of 0.5 at 600 nm was reached (equivalent to 1 × 10^7^ cells/mL). Three serial dilutions were prepared and 5 µL of each was spotted onto a plate (approximately 1 × 10^5^, 1 × 10^4^, 1 × 10^3^, and 1 × 10^2^ cells/spot). YPAD plates were supplemented with 200 µg/mL of each Schiff base, and one plate supplemented with 2% DMSO and one plate with YPAD without any drug were used as controls. Plates were incubated at 25 °C for 24, 48, and 72 h.

### 4.7. DPPH Radical Scavenging Assay

The stock solutions of the compounds were prepared by dissolving **3** in DMSO to a concentration of 4 mg/mL, then diluted with methanol to give a concentration of 400 µg/mL. This was used immediately. The experimental procedure was adapted from the literature [59]. Briefly, in a 96-well plate, 100 µL of solution of DPPH (2, 2-diphenyl-1-picrylhydrazyl) radical (0.2 mM in methanol) was added to 100 µL of methanolic solutions of **3a**–**j**, **4**, or **5**, prepared as serial two-fold dilutions from the stock solution. Standards were also prepared in the same concentrations. The mixture was incubated in the dark at room temperature for 30 min, and the absorbance was read at 515 nm on a microplate reader, a BioTek Synergy H1 spectrophotometer. 

The percentage of DPPH scavenging was then calculated by using the following formula:$$\%\;\mathrm{DPPH}\;\mathrm{scavenging}=100*\left[\frac{\left(A_{\mathrm{sample}+\mathrm{DPPH}}-A_{\mathrm{sample}\;\mathrm{blank}}\right)}{\left(A_{\mathrm{DPPH}}-A_{\mathrm{solvent}}\right)}\right]$$

The antioxidant activity of the compound was expressed as IC_50_, which is defined as the concentration that could scavenge 50% of the DPPH free radicals.

### 4.8. Theoretical Prediction of ADME Properties and Bioactivity Scores

The predictions of ADME toxicity parameters and bioactivity scores were calculated as described previously [49,50,60,61]. The Osiris DataWarrior version 4.4.4 software on a Windows 10 operating system [50] was used to determine the drug-like properties of the synthesized compounds. These properties included the molecular polar surface area (PSA), octanol–water partition coefficient (clogP), aqueous solubility (clogS), number of rotatable bonds (nrotb), number of hydrogen donors (HBD), number of hydrogen acceptors (HBA), toxicities (mutagenic, tumorigenic, irritant, and reproductive), drug scores (DS), and drug likeness (DL). The bioactivity scores, including activities as a GPCR ligand, ion channel modulator, kinase inhibitor, nuclear receptor ligand, protease inhibitor, and enzyme inhibitor, were calculated using the Molinspiration Cheminformatics software.

## Figures and Tables

**Figure 1 molecules-24-02696-f001:**
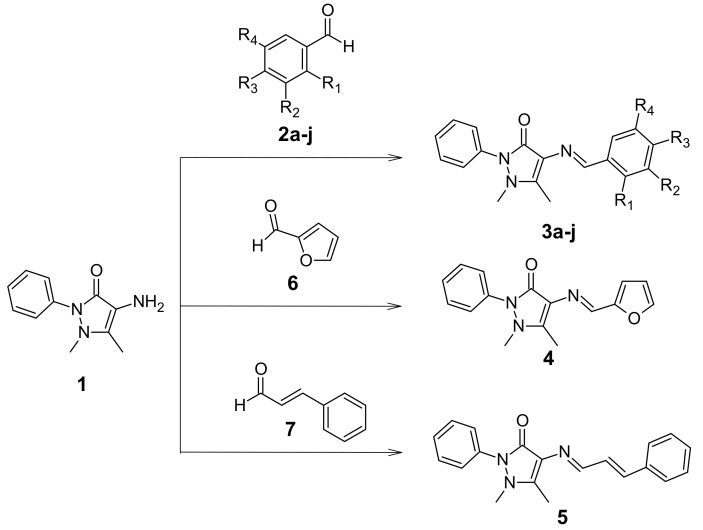
Synthesis of Schiff base derivatives of 4-aminoantipyrine. Depiction of the scheme followed to synthesize the Schiff bases described in this study, designated as **3a**–**j**, **4**, and **5**.

**Figure 2 molecules-24-02696-f002:**
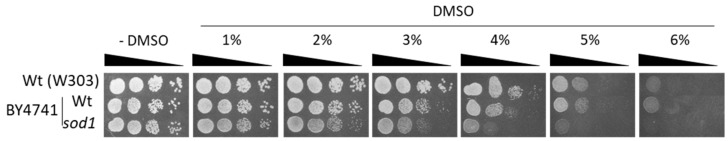
DMSO toxicity. Drop test of serial dilution of yeast cell cultures (wild type from genetic background W303, wild type from genetic background BY4741, and *sod1* mutant from genetic background BY4741) exposed to different DMSO concentrations ranging from 1 to 6%. A control without DMSO (−DMSO) is included.

**Figure 3 molecules-24-02696-f003:**
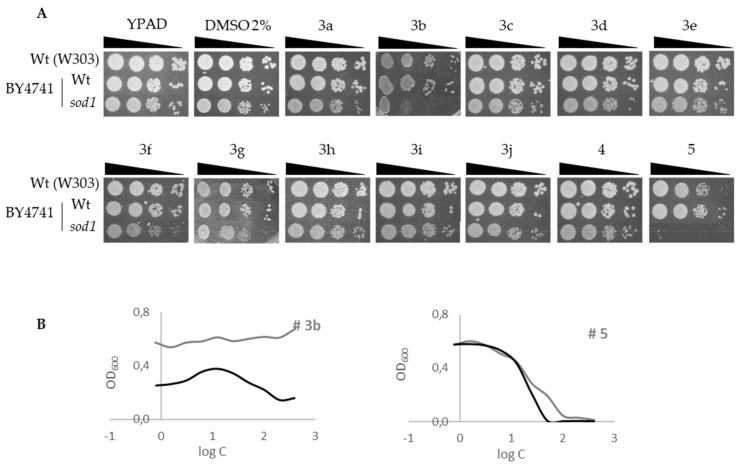
Oxidative effect. (**A**) Drop test of serial dilution of yeast cell cultures (wild type from genetic background W303, wild type from genetic background BY4741, and *sod1* mutant from genetic background BY4741) exposed to Schiff bases at 200 µg/mL. A control without drug and with 2% DMSO are included. (**B**) Graphic representation of optical density (OD600) of wild-type (black line) and *sod1* (grey line) cells (BY4741) exposed to increasing concentrations of compounds **3b** and **5**.

**Figure 4 molecules-24-02696-f004:**
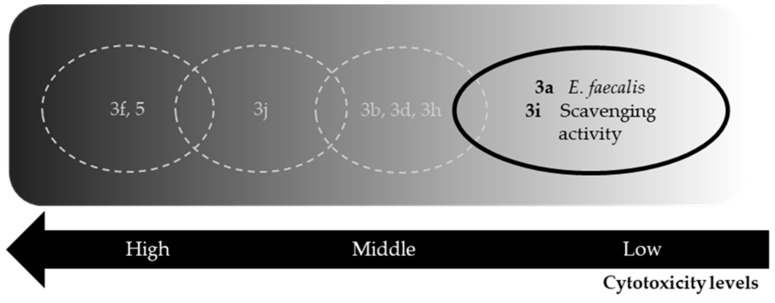
Summarizing diagram highlighting the evaluated Schiff bases. Relevant compounds are in black/bold track, non-relevant compounds discarded because of associated cytotoxicity are in light grey/dashed tracks. A grey scale indicates the intensity of cytotoxicity.

**Table 1 molecules-24-02696-t001:** Description of substituents R_1_, R_2_, R_3_, and R_4_ and the obtained synthesis yield (%) of Schiff bases.

Compounds	R_1_	R_2_	R_3_	R_4_	Yield (%)
**3a**	H	H	H	H	77
**3b**	OH	H	H	H	94
**3c**	H	H	OMe	H	86
**3d**	H	OMe	OH	H	94
**3e**	H	OH	OMe	H	93
**3f**	OH	H	H	NO_2_	85
**3g**	H	NO_2_	H	H	95
**3h**	H	H	N(Me)_2_	H	95
**3i**	H	OH	OH	H	97
**3j**	H	OMe	OH	OMe	91
**4**	-	-	-	-	86
**5**	-	-	-	-	90

**Table 2 molecules-24-02696-t002:** MIC, IC_50_, and NIC values determined for DMSO in the indicated microorganism.

	*A. niger*	*C. albicans*	*S. cerevisiae*	*E. faecalis*	*E. coli*	*S. aureus*
**MIC** **DMSO (%) IC_50_** **NIC**	19.22 ± 1.3 4.36 ± 2.0 1.78 ± 0.4	14.98 ± 0.6 8.50 ± 0.5 4.94 ± 0.5	71.47 ± 1.7 35.62 ± 1.0 5.04 ± 1.1	20.02 ± 1.3 8.66 ± 0.6 3.68 ± 0.2	13.81 ± 3.5 16.36 ± 0.33 11.95 ± 2.9	32.64 ± 6.9 20.36 ± 5.2 8.52 ± 0.9

**Table 3 molecules-24-02696-t003:** Growing inhibitory parameters (MIC, NIC, and IC_50_, expressed in μg/mL), determined by serial microdilution in 96-well plates, for Schiff bases **3a**–**j**, **4**, and **5**. The sensitivity was tested for the microorganism indicated in the first column. Geneticin and voriconazole were used as control against fungi, ciprofloxacin for bacteria, and amphotericin and pentamidine for leishmania.

	3a	3b	3c	3d	3e	3f	3g	3h	3i	3j	4	5	GN	CP	VR	AP	PT
**MIC** ***A. niger* IC_50_** **NIC**	NE	NE	NE	NE	NE	32.6 ± 2.1 10.6 ± 0.8 3.4 ± 0.5	NE	NE	NE	NE	NE	NE	ND	NA	1.2 ± 0.05 0.2 ± 0.02 0.05 ± 0.01	NA	NA
**MIC** ***C. albicans* IC_50_** **NIC**	NE	13.2 ± 1.6 5.8 ± 2.5 3.0 ± 2.2	NE	NE	NE	15.6 ± 0.9 10.4 ± 0.6 7.5 ± 1.1	NE	NE	NE	NE	NE	NE	ND	NA	29.7 ± 3.52 0.9 ± 0.07 0.00 ± 0.00	NA	NA
**MIC** ***S. cerevisiae* IC_50_** **NIC**	NE	NE	NE	NE	NE	55.9 ± 6.2 17.9 ± 1.4 5.6 ± 0.9	179.2 ± 59.5 105.1 ± 22.1 63.7 ± 8.7	NE	NE	NE	NE	76.7 ± 5.7 22.8 ± 0.3 6.6 ± 0.7	93.7 ± 0.9 59.9 ± 0.9 44.5 ± 1.7	NA	NA	NA	NA
**MIC** ***E. faecalis* IC_50_** **NIC**	32.3 ± 11.7 12.2 ± 3.3 3.6 ± 0.7	NE	NE	NE	NE	NE	NE	NE	NE	NE	NE	NE	NA	1.15 ± 0.42 0.49 ± 0.07 0.30 ± 0.07	NA	NA	NA
**MIC** ***S. aureus* IC_50_** **NIC**	NE	NE	NE	NE	NE	27.7 ± 4.7 9.5 ± 1.5 3.3 ± 1.5	NE	NE	NE	NE	NE	NE	NA	0.44 ± 0.09 0.10 ± 0.01 0.06 ± 0.02	NA	NA	NA
**MIC** ***E. coli* IC_50_** **NIC**	NE	49.8 ± 5.0 12.1 ± 0.5 2.8 ± 0.5	NE	10.2 ± 0.0 3.3 ± 0.8 2.3 ± 0.1	NE	47.7 ± 1.0 15.8 ± 0.1 4.9 ± 0.2	NE	NE	NE	NE	NE	22.6 ± 14.1 12.4 ± 6.2 5.1 ± 2.4	NA	0.15 ± 0.009 0.07 ± 0.004 0.06 ± 0.003	NA	NA	NA
**MIC** ***P. aeruginosa* IC_50_** **NIC**	NE	NE	NE	NE	NE	NE	NE	NE	NE	NE	NE	NE	NA	0.47 ± 0.1 0.94 ± 0.0 ND	NA	NA	NA
**MIC** ***L. mexicana* IC_50_ *** **NIC**	NE	NE	NE	NE	NE	5.3 ± 2.8	NE	NE	NE	NE	NE	81.5 ± 12.5	NA	NA	NA	0.18 ± 0.07	4.50 ± 0.46

* Leishmanicidal activity expressed in µM; MIC: minimum inhibitory concentration; NIC: noninhibitory concentration; IC_50_: inhibitory concentration of 50% of population. NA: Not applicable; ND: not determined; NE: no effect. GN: geneticin; CP: ciprofloxacin; VR: voriconazole; AP: amphotericin; PT: pentamidine.

**Table 4 molecules-24-02696-t004:** Antioxidant activities of Schiff bases. DPPH scavenging IC_50_ values (μM), for Schiff bases **3a**–**j**, **4**, and **5**. Ascorbic acid, quercetin, and caffeic acid were used as controls for antioxidant activity.

Entry	Compound	DPPH Scavenging Activity IC_50_ (µM)
**1**	**3a**	>200
**2**	**3b**	NA
**3**	**3c**	>200
**4**	**3d**	28.33 ± 4.35
**5**	**3e**	>200
**6**	**3f**	>200
**7**	**3g**	>200
**8**	**3h**	129.4 ± 18.7
**9**	**3i**	18.9 ± 2.4
**10**	**3j**	15.7 ± 3.2
**11**	**4**	>200
**12**	**5**	>200
**13**	ascorbic acid	14.5 ± 2.2
**14**	quercetin	7.3 ± 1.0
**15**	caffeic acid	16.2 ± 2.4

DPPH: *N*,*N*-diphenyl-*N*′-picrylhydrazyl.

**Table 5 molecules-24-02696-t005:** MIC, IC_50_, and NIC values obtained for wild type and *sod1* strains exposed to compounds **3b** and **5**.

		Wt	*sod1*
***3b***	**MIC** **IC_50_** **NIC**	208.2 ± 4.0 192.0 ± 0.6 178.0 ± 2.1	164.1 ± 0.1 100.7 ± 4.2 53.3 ± 0.1
***5***	**MIC** **NIC IC_50_**	65.2 ± 0.2 22.6 ± 0.6 8.1 ± 0.4	39.4 ± 0.9 19.5 ± 0.8 10.4 ± 1.1

**Table 6 molecules-24-02696-t006:** Cytotoxicity of the investigated compounds. Cytotoxicity is expressed as % of live cells after exposure to 400 µM of Schiff bases (compounds with limited solubility were evaluated at 300 µM). Equivalent concentrations are expressed in mg/mL units for comparison to the IC_50_ of saponine (in mg/mL), used as control.

Entry	Compound	% Live Cells	µM	mg/mL
**1**	**3a**	27.1 ± 3.4	400	0.1165
**2**	**3b**	19.3 ± 0.2	400	0.1229
**3**	**3c**	26.1 ± 1.9	400	0.1285
**4**	**3d**	17.6 ± 0.4	400	0.1349
**5**	**3e**	27.1 ± 0.8	300 *	0.1012
**6**	**3f**	4.5 ± 1.4	400	0.1409
**7**	**3g**	22.1 ± 0.6	300 *	0.1009
**8**	**3h**	20.4 ± 1.7	300 *	0.1003
**9**	**3i**	34.4 ± 2.4	400	0.1293
**10**	**3j**	14.2 ± 0.6	400	0.1470
**11**	**4**	23.8 ± 3.4	400	0.1125
**12**	**5**	2.7 ± 0.5	400	0.1270
**13**	**Saponine** **	IC_50_ (50%)	NA	0.1410 ± 0.02

* Compounds with limited solubility were evaluated at 300 µM. ** The saponine used was a commercial mix of different saponines, so the µM cannot be calculated.

**Table 7 molecules-24-02696-t007:** Theoretical prediction of adsorption, distribution, metabolism, and excretion (ADME) properties of Schiff bases **3a**–**j**, **4**, and **5**, calculated using Osiris DataWarrior software. Voriconazol and ciprofloxacin were included as controls.

Compound	MW	HBA	HBD	nrotb	PSA	M	T	R	I	cLogP	cLogS	DL	DS
**3a**	291.35	4	0	3	35.91	high	NE	NE	NE	2.16	−3.24	4.25	0.52
**3b**	307.35	5	1	3	56.14	high	NE	NE	NE	1.81	−2.94	4.25	0.53
**3c**	321.38	5	0	4	45.14	high	NE	NE	NE	2.09	−3.26	4.24	0.51
**3d**	337.38	6	1	4	65.37	high	low	NE	NE	1.74	−2.96	4.24	0.41
**3e**	337.38	6	1	4	65.37	high	NE	NE	NE	1.74	−2.96	4.24	0.52
**3f**	352.35	8	1	4	101.96	high	NE	NE	NE	0.67	−3.40	−0.85	0.33
**3g**	336.35	7	0	4	81.73	high	NE	NE	NE	1.01	−3.70	−0.85	0.32
**3h**	334.42	5	0	4	39.15	high	high	NE	NE	2.05	−3.28	4.56	0.30
**3i**	323.35	6	2	3	76.37	high	NE	NE	NE	1.47	−2.65	4.25	0.53
**3j**	367.40	7	1	5	74.60	high	NE	NE	NE	1.67	−2.98	4.24	0.50
**4**	281.31	5	0	3	49.05	high	NE	NE	NE	1.35	−2.92	4.08	0.54
**5**	317.39	4	0	4	35.91	high	NE	NE	NE	2.18	−3.53	4.28	0.50
**VR**	349.32	6	1	5	76.72	NE	NE	NE	NE	1.48	−3.23	4.08	0.84
**CP**	331.35	6	2	3	72.88	NE	NE	NE	NE	−1.53	−3.32	2.05	0.82

MW: molecular weight; HBA: number of hydrogen bond acceptors; HBD: number of hydrogen bond donors; nrotb: number of rotatable bonds; PSA: polar surface area; M: mutagenicity; T; tumorigenicity; R: reproductive effect; I: irritant effect; cLogP: logarithm of compound partition coefficient between n-octanol and water; cLogS: logarithm of compound aqueous solubility; DL: drug-likeness; DS: drug score. VR: voriconazole; CP: ciprofloxacin. NE: no effect.

**Table 8 molecules-24-02696-t008:** Predicted bioactivity scores of the synthesized Schiff bases **3a**–**j**, **4**, and **5**, calculated using Molinspiration Cheminformatics Osiris software. Voriconazol and ciprofloxacin were included as controls.

Entry	Compound	GPCR	ICM	KI	NRL	PI	EI
**1**	**3a**	−0.90	−1.12	−0.63	−1.08	−1.07	−0.53
**2**	**3b**	−0.82	−1.14	−0.58	−0.92	−0.95	−0.48
**3**	**3c**	−0.85	−1.10	−0.60	−0.98	−1.00	−0.54
**4**	**3d**	−0.79	−1.03	−0.54	−0.90	−0.98	−0.47
**5**	**3e**	−0.79	−1.03	−0.54	−0.90	−0.98	−0.47
**6**	**3f**	−0.88	−1.07	−0.65	−0.91	−0.96	−0.55
**7**	**3g**	−0.93	−1.05	−0.69	−1.03	−1.05	−0.59
**8**	**3h**	−0.76	−1.01	−0.51	−0.91	−0.92	−0.48
**9**	**3i**	−0.77	−1.01	−0.54	−0.86	−0.95	−0.44
**10**	**3j**	−0.75	−0.96	−0.49	−0.86	−0.89	−0.41
**11**	**4**	−1.01	−1.26	−0.96	−1.35	−1.30	−0.70
**12**	**5**	−0.63	−0.84	−0.75	−0.93	−0.96	−0.51
**13**	**VRC**	0.23	0.17	0.14	−0.22	0.02	0.19
**14**	**CP**	0.12	−0.04	−0.07	−0.19	−0.20	0.28

GPCR: GPCR ligand; ICM: ion channel modulator; KI: kinase inhibitor; NRL: nuclear receptor ligand; PI: protease inhibitor; EI: enzyme inhibitor; VR: voriconazole; CP: ciprofloxacin.

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
