# Peer review of "Characterization of Antimicrobial, Antioxidant, and Leishmanicidal Activities of Schiff Base Derivatives of 4-Aminoantipyrine"

_molecules, 2019, doi:10.3390/molecules24152696_

Reviewer 1 Report

Dear Authors,

The manuscript presented the synthesis and very interesting biological studies of the Schiff bases.

As for synthesis and descriptions, they are correct.

The biological studies is a reach.

As for organization of the manuscript. I did not find the Conclusion. So, the Conclusion should be added.

My opinion, the manuscript can be published in Molecules.

Minor correction.

338 “The hydroxyl group in the  ortho  position can form an intermolecular hydrogen bond with the imine nitrogen [47] …” - intermolecular must be intramolecular hydrogen bond, which is a strong intramolecular hydrogen bond see papers by Filarowski et al. especially, review in MDPI Molecules (Molecules 2016, 21(12), 1657 Some Brief Notes on Theoretical and Experimental Investigations of Intramolecular Hydrogen Bond), which is more appropriate than cited 47 paper and should be cited.

Sincerely

Author Response

Response to Reviewer 1 Comments

First, we want to thank the reviewer for his/her positive comments and suggestions. Modifications in the text are highlight with track changes mode in MS word. Specific changes to our manuscript are listed below:

Open Review

English language and style

( ) Extensive editing of English language and style required 
( ) Moderate English changes required 
( ) English language and style are fine/minor spell check required 
(x) I don't feel qualified to judge about the English language and style 

Yes

Can be improved

Must be improved

Not applicable

Does the introduction provide sufficient background and include   all relevant references?

(x)

( )

( )

( )

Is the research design appropriate?

( )

(x)

( )

( )

Are the methods adequately described?

(x)

( )

( )

( )

Are the results clearly presented?

(x)

( )

( )

( )

Are the conclusions supported by the results?

( )

(x)

( )

( )

Comments and Suggestions for Authors

Dear Authors,

The manuscript presented the synthesis and very interesting biological studies of the Schiff bases.

As for synthesis and descriptions, they are correct.

The biological studies is a reach.

As for organization of the manuscript. I did not find the Conclusion. So, the Conclusion should be added.

My opinion, the manuscript can be published in Molecules.

Minor correction.

Point 1: 338 “The hydroxyl group in the  ortho  position can form an intermolecular hydrogen bond with the imine nitrogen [47] …” - intermolecular must be intramolecular hydrogen bond, which is a strong intramolecular hydrogen bond see papers by Filarowski et al. especially, review in MDPI Molecules (Molecules 2016, 21(12), 1657 Some Brief Notes on Theoretical and Experimental Investigations of Intramolecular Hydrogen Bond), which is more appropriate than cited 47 paper and should be cited.

Response 1: Yes, you are right, it is an intramolecular hydrogen bond. We corrected the mistake and changed the reference as you suggest.

Sincerely

Submission Date

26 March 2019

Date of this review

29 Mar 2019 11:01:50

Reviewer 2 Report

The first sentence in the abstract concerns literature background, not the highlights from Authors’ results. Similar notice for beginning of section 3. Discussion.

The manuscript should be checked for minor errors, e.g.:

i) page 3: some compounds’ numbers without bold,

ii) line 255, 287, 567 (are abbreviations HBD, HBA hydrogen bond donors / acceptors?),

iii) line 399: which “others fields” the Authors think about?

iv) literature section: not all bibliographic data or provided variously or with typo errors, e.g. reference 1, 7, 13, 21, 27, 29, 31, 44.

Author Response

Response to Reviewer 2 Comments

We are also grateful to Reviewer 2 and Editor for their positive comments.

Open Review

English language and style

( ) Extensive editing of English language and style required 
( ) Moderate English changes required 
( ) English language and style are fine/minor spell check required 
(x) I don't feel qualified to judge about the English language and style 

Yes

Can be improved

Must be improved

Not applicable

Does the introduction provide sufficient background and include   all relevant references?

(x)

( )

( )

( )

Is the research design appropriate?

(x)

( )

( )

( )

Are the methods adequately described?

(x)

( )

( )

( )

Are the results clearly presented?

(x)

( )

( )

( )

Are the conclusions supported by the results?

(x)

( )

( )

( )

Comments and Suggestions for Authors

Point 1: The first sentence in the abstract concerns literature background, not the highlights from Authors’ results. Similar notice for beginning of section 3. Discussion.

Response 1:  The suggestion is accepted and included in the new title.

The manuscript should be checked for minor errors, e.g.:

Point 2: i) page 3: some compounds’ numbers without bold.

Response 2: Mistakes are reviewed and corrected. We apologize for these errors, apparently, we lost the format when translating the text to the template.

Point 3: ii) line 255, 287, 567 (are abbreviations HBD, HBA hydrogen bond donors / acceptors?),

Response 3: Yes, HBD is the number of hydrogen bond donor and HBA is the number of hydrogen bond acceptors. The description about this was missing in the text, apparently we lost it when translating the text to the template. We added that information in line 202.

Point 4: iii) line 399: which “others fields” the Authors think about?

Response 4: We were thinking in the potential antioxidant activity, with potential application as preservative (foods, drugs in solution). However, we agree that the redaction is not as clear as we though, since the main application of these compounds is therapy, and we wanted to distinguish between antimicrobial activities from other described for Schiff bases (anti diabetic, antitumor, analgesic, among others). The text has been modified.

Point 5: iv) literature section: not all bibliographic data or provided variously or with typo errors, e.g. reference 1, 7, 13, 21, 27, 29, 31, 44.

Response 5: We check all the references and correct them to the right format.

Submission Date

26 March 2019

Date of this review

01 Apr 2019 16:09:55